# A topological fluctuation theorem

Benoît Mahault [1], Evelyn Tang[1] & Ramin Golestanian [1,2 ✉]

Fluctuation theorems specify the non-zero probability to observe negative entropy production, contrary to a naive expectation from the second law of thermodynamics. For closed particle trajectories in a fluid, Stokes theorem can be used to give a geometric characterization of the entropy production. Building on this picture, we formulate a topological fluctuation theorem that depends only by the winding number around each vortex core and is insensitive to other aspects of the force. The probability is robust to local deformations of the particle trajectory, reminiscent of topologically protected modes in various classical and quantum systems. We demonstrate that entropy production is quantized in these strongly fluctuating systems, and it is controlled by a topological invariant. We demonstrate that the theorem holds even when the probability distributions are non-Gaussian functions of the generated heat.

[1] Max Planck Institute for Dynamics and Self-Organization, 37077 Göttingen, Germany. [2] Rudolf Peierls Centre for Theoretical Physics, University of Oxford, Oxford OX1 3PU, UK. ✉email: ramin.golestanian@ds.mpg.de

Concepts from topology have played a key role in understanding a wide range of physical phenomena by providing an intuitive and mathematically rich effective description of the system[1,2]. In classical systems, topology has played a crucial role, for instance, in characterizing 2D turbulence[3] and defect-mediated phase transitions[4]. Topological defects abound in soft matter systems, such as dislocation and disclination pairs in 2D melting[5] and various liquid crystalline systems[6], and have recently featured in a variety of intrinsically non-equilibrium active matter systems[7–11]. An early example of topological protection arose from use of the Gauss-Bonnet theorem in the physics of membranes[12]. More recent work examined relevant topological invariants in mechanical lattices[13] or dissipative systems in continuous space[14] or lattice models with underlying periodic structure[15–20]. Many such systems exhibit non-Hermitian properties such as exceptional points[21–25] and a non-zero topological vorticity of the edge state[14,20]. The topological systems that support protected edge states are robust to disorder and perturbations, providing a key towards understanding such phenomena.

Fluctuations can give rise to particle trajectories with negative entropy, which appears to contradict a fundamental law of macroscopic physics. Fluctuation theorems provide a quantitative, probabilistic prediction for this negative entropy production. These universal laws are valid even during processes that drive systems far from equilibrium. While the probability to observe such "violations" is typically exponentially small in the relevant system size, it can be appreciable in small systems. The first theorem was discovered over two decades ago by computer simulations and justified heuristically[26], then proven for a large class of systems[27–29]. This led to a class of relations dealing with the distribution functions of thermodynamic quantities such as exchanged heat, applied work or entropy production[30]. Additional relations were moreover found in a variety of systems with non-reversible microscopic dynamics[31–36]. Further work extended the concepts of thermodynamics to the level of individual trajectories[37,38]. Another key insight related entropy production in the medium to that part of the stochastic action, which determines the weight of trajectories that is odd under time reversal[39,40]. These and other developments opened the possibility for experimental or numerical measurements on the single molecule level, providing verification of the fluctuation theorems[41,42]. Notably, critical insights were obtained for the behavior of bio-molecules[43]. Brownian dynamics of tracers in the presence of vortex-like singularities has been studied with an empasis on the winding number distribution and extensions to entanglement problems in polymer physics[44–48]. However, no connection has so far been made to stochastic thermodynamics.

Here, we identify a topological invariant that predicts observable quantities, in a strongly fluctuating system without underlying periodic structure. Our analysis allows us to quantify the ratio of particles with negative entropy production, purely as a function of winding number around vortex cores. We build on the geometrical properties of individual trajectories under the influence of external forces. This is studied in the context of a stochastic particle in a force-field, which is ubiquitous in nature (see Fig. 1). In particular, a particle moving around a closed path picks out only the non-conservative component of the force-field, which gives the entropy production. This allows us to formulate a topological fluctuation theorem based only on the vortex winding number. While previous work looked at entropy production in flow-fields[49,50], the topological equivalence of particle trajectories and its consequences have not been studied. We begin with a general statement of the theorem using the examination of entropy production in the medium. We then demonstrate it in various examples including one or several vortices by calculating

the corresponding exact winding number distribution or performing Brownian dynamics simulations. We find that even when the winding number distributions are non-Gaussian functions of the vortex circulation, the theorem holds exactly.

## Results

Consider the motion in $d$ dimensions of a tracer particle with diffusivity $D$ and mobility $\mu$ in a stationary force field $\mathbf{F}(\mathbf{r})$, where $D = \mu k_B T$ with $k_B T$ representing the thermal energy. The stochastic dynamics of the particle is characterized as a function of time $0 \leq \tau \leq t$ by its trajectory $\mathbf{r}(\tau)$, which satisfies the Langevin equation

$$\dot{\mathbf{r}}(\tau) = \mu \mathbf{F}(\mathbf{r}(\tau)) + \sqrt{2D}\,\boldsymbol{\xi}(\tau), \qquad (1)$$

where $\boldsymbol{\xi}$ represents a $d$-dimensional Gaussian white noise of zero mean and unit variance. Using the Helmholtz-Hodge decomposition, the force field can generally be separated into a conservative component that derives from a potential $U(\mathbf{r})$ and a rotational component $\mathbf{f}(\mathbf{r})$ that satisfies $\nabla \cdot \mathbf{f}(\mathbf{r}) = 0$, i.e., $\mathbf{F}(\mathbf{r}) = -\nabla U(\mathbf{r}) + \mathbf{f}(\mathbf{r})$. Let us assume that the rotational component of the force field is generated by a topological defect, such as the vortex line shown in Fig. 1a for the physically relevant case $d = 3$. In the general case, such a defect takes the form of codimension 2 manifold $\Sigma$ giving rise to a vorticity field satisfying

$$Q \equiv \int_S dr^i \wedge dr^j \left( \partial_i f_j - \partial_j f_i \right), \qquad (2)$$

where $S$ is an arbitrary smooth surface intersecting $\Sigma$ transversally at a single point, the constant $Q$ is the strength of the vortex and summation is performed over $i < j$ (distinct pairs)[51,52]. For such a vortex-induced force-field, we derive a topological fluctuation theorem in terms of the probability $p(n, t)$ for any closed trajectory of duration $t$ to wind $n$ times around the vortex, which reads

$$\frac{p(-n, t)}{p(n, t)} = \exp\left( -\frac{nQ}{k_B T} \right), \qquad (3)$$

for any $U(\mathbf{r})$ (see Fig. 1a for a depiction of a representative force profile and closed loop in $d = 3$). Here, $Q$ corresponds to the quantum of heat generated via the closed trajectories of the tracer particle that enclose the vortex domain. It is helpful to define $\gamma \equiv Q/(k_B T)$ as the quantum of entropy production (in units of $k_B$). Figure 1b shows a number of exemplar closed trajectories with different winding numbers. Note that the theorem [Eq. (3)] is valid at any time $t$. Therefore, while the positive and negative winding number distributions are expected to evolve with time and be affected by conservative contributions from the force, their ratio is topologically protected.

The derivation of the topological fluctuation theorem starts from the probability distribution of the particle to be at position $\mathbf{x}$ after time $t$ starting initially from $\mathbf{x}_0$, which can be be found via

$$\mathcal{P}(\mathbf{x}, t | \mathbf{x}_0, 0) = \int_{\mathbf{r}(0) = \mathbf{x}_0}^{\mathbf{r}(t) = \mathbf{x}} \mathcal{D}\mathbf{r}(\tau) P[\mathbf{r}(\tau) | \mathbf{x}_0], \qquad (4)$$

where the probability of a specific stochastic trajectory is defined in terms of the Onsager-Machlup action associated with Eq. (1) as

$$P[\mathbf{r}(\tau)|\mathbf{x}_0] =$$
$$\mathcal{N} \exp\left( -\int_0^t d\tau \left[ \frac{1}{4D} \left( \dot{\mathbf{r}}(\tau) - \mu \mathbf{F}(\mathbf{r}(\tau)) \right)^2 + \frac{\mu}{2} \nabla \cdot \mathbf{F}(\mathbf{r}(\tau)) \right] \right), \qquad (5)$$

where $\mathcal{N}$ is a normalization factor, and the Stratonovich convention is implied. Defining the kinematically reversed, or backward, trajectory $\tilde{\mathbf{r}}(\tau)$ as $\tilde{\mathbf{r}}(\tau) \equiv \mathbf{r}(t - \tau)$, we then find from Eq. (5) that the ratio between the probabilities of the backward and forward paths is given by the part of the stochastic action that

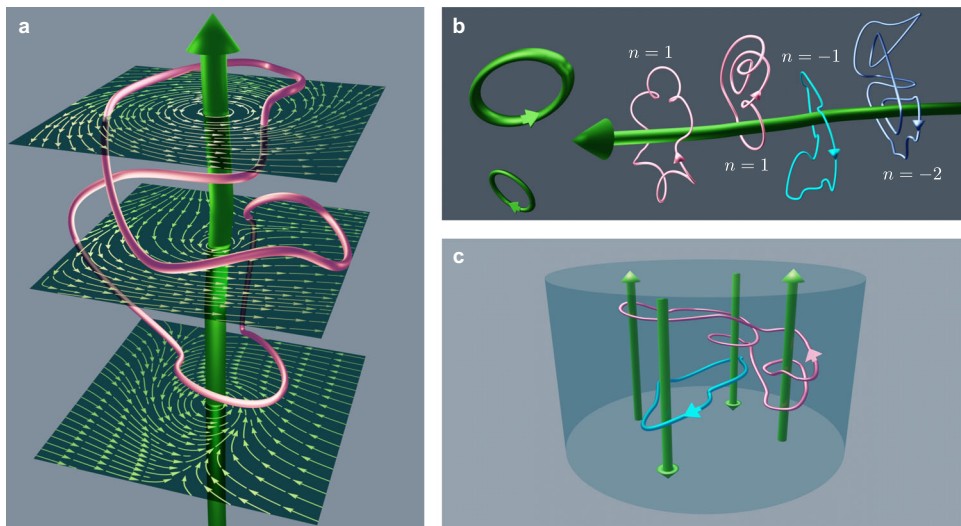

**Fig. 1 Vizualizations of tracer particle trajectories in vortex force-fields. a** A tracer particle undergoes diffusion and drift in a force-field whose non-conservative component is generated by a vortex line (thick green), tracing out a stochastic trajectory (thin irregular loop). The force-field has the same rotational component throughout but varying irrotational components (the cross-sections depict force-field streamlines)—however, closed loops pick up only the rotational component. **b** Different particle trajectories around a single vortex line (green) can be characterized by the winding number $n$. The leftmost two are topologically equivalent with $n = 1$, while the rightmost two have the opposite sign and winding numbers $n = -1$ and $n = -2$, respectively. **c** Force-fields can contain multiple vortex-cores (thick green lines or rings), around which particles can trace closed trajectories (thin irregular loops).

is odd under this transformation, namely,

$$\frac{P[\tilde{\mathbf{r}}(\tau)|\mathbf{x}]}{P[\mathbf{r}(\tau)|\mathbf{x}_0]} = \exp(-\Gamma), \qquad (6)$$

where $\Gamma \equiv \frac{1}{k_B T} \int_0^t d\tau \, \dot{\mathbf{r}}(\tau) \cdot \mathbf{F}(\mathbf{r}(\tau))$ corresponds to the heat generated (in units of $k_B T$), or entropy production, during the motion of the particle along the trajectory. We then consider a closed trajectory described by a curve $C$ (by setting $\mathbf{x} = \mathbf{x}_0$), which does not intersect with $\Sigma$, and implement the Stokes Theorem $\int_{\partial S} \omega = \int_S d\omega$ for the 1-form $\omega = F_i dr^i$ over the surface $S$ enclosed by $C^2$, namely

$$\begin{aligned}\Gamma &= \frac{1}{k_B T} \oint_C d\mathbf{r} \cdot \mathbf{F}(\mathbf{r}) = \frac{1}{k_B T} \int_{\partial S = C} dr^i F_i \\ &= \frac{1}{k_B T} \int_S dr^i \wedge dr^j (\partial_i F_j - \partial_j F_i).\end{aligned} \qquad (7)$$

It is manifest in Eq. (7) that $\Gamma$ picks out only the rotational component of the force. Moreover, using the decomposition $\mathbf{F} = -\nabla U + \mathbf{f}$ and Eq. (2) directly yields $\Gamma = \gamma n$, with $n$ the number of times that the forward trajectory winds around the vortex singularity. We thus find that $\Gamma$ is independent of the initial position $\mathbf{x}_0$ or the specific shape of $C$ so long as it corresponds to the same winding number $n$. Hence, $\Gamma$ is identical for topologically equivalent curves (see Fig. 1b), and independent of the conservative component of the force-field.

To complete the calculation, we thus set $\mathbf{x} = \mathbf{x}_0$ in Eq. (6) and integrate over all initial positions and topologically equivalent loops to obtain a topological fluctuation theorem for vortex-induced force fields (details in Methods)

$$\frac{p(-n, t)}{p(n, t)} = \exp(-n\gamma), \qquad (8)$$

where $p(n, t)$ denotes the probability for any closed trajectory of length $t$ to wind $n$ times around the vortex axis.

The above derivation is easily generalizable to the case of $m$ non-intersecting vortex branes. Denoting $\gamma_i$ and $n_i$ the dimensionless quantum of heat and winding number associated with a given vortex $i$, the entropy production becomes $\Gamma = \sum_{i=1}^m \gamma_i n_i$. Integrating Eq. (6) over initial conditions and trajectories, which

share the same set of winding numbers $\{n_i\}_{i=1,\dots m}$, the topological fluctuation theorem for multiple vortices then reads

$$\frac{p(\{-n_i\}, t)}{p(\{n_i\}, t)} = \exp\left(-\sum_{i=1}^m \gamma_i n_i\right), \qquad (9)$$

where $p(\{n_i\}, t)$ is the probability that a closed trajectory of length $t$ has wrapped $n_i$ times around each vortex $i$ for $i = 1, \dots m$.

As for a single vortex, Eq. (9) is set by topology and thus holds at all times and for any conservative force affecting the particle motion. Moreover, as both sides of Eq. (9) depend only on the set of winding numbers $\{n_i\}$ reached at time $t$, the fluctuation theorem is insensitive to the history leading to $\{n_i\}$ and in particular to the order in which the trajectory winds around each vortex. Since any combination of winding numbers leading to the same value of $\Gamma$ leaves the r.h.s. of Eq. (9) unchanged, the latter can be further summed over all such configurations in order to get a weaker formulation of the theorem:

$$\frac{p(\Gamma = -\sum_{i=1}^m \gamma_i n_i, t)}{p(\Gamma = \sum_{i=1}^m \gamma_i n_i, t)} = \exp\left(-\sum_{i=1}^m \gamma_i n_i\right). \qquad (10)$$

**Exact solution for a single vortex.** To shed some light on the topological fluctuation theorem (3), we now focus on the physically relevant case $d = 3$ and consider the motion of a tracer particle in a flow field created by a single straight vortex line oriented along $\mathbf{e}_z$, which is given in cylindrical coordinates $(r, \phi, z)$ by

$$\mathbf{f}(\mathbf{r}) = \frac{Q}{2\pi r} \mathbf{e}_\phi, \qquad \nabla \times \mathbf{f}(\mathbf{r}) = Q\mathbf{e}_z \delta^2(\mathbf{r}_\perp), \qquad (11)$$

where $\mathbf{e}_\phi = (-\sin\phi, \cos\phi, 0)$ and $\mathbf{r}_\perp = r(\cos\phi, \sin\phi)$, so that from Eq. (7) we can readily evaluate $\Gamma = \frac{1}{k_B T} \int_S d\mathbf{S} \cdot \nabla \times \mathbf{F}(\mathbf{r}) = \gamma n$. The simple case where the particle motion is restricted to a ring following the flow streamlines, such that the drive is effectively uniform, has been treated previously[38]. This problem (generalizable to higher dimensions) corresponds to a biased random walk and the fluctuation theorem can be shown to result from the Gaussian form of the winding distribution.

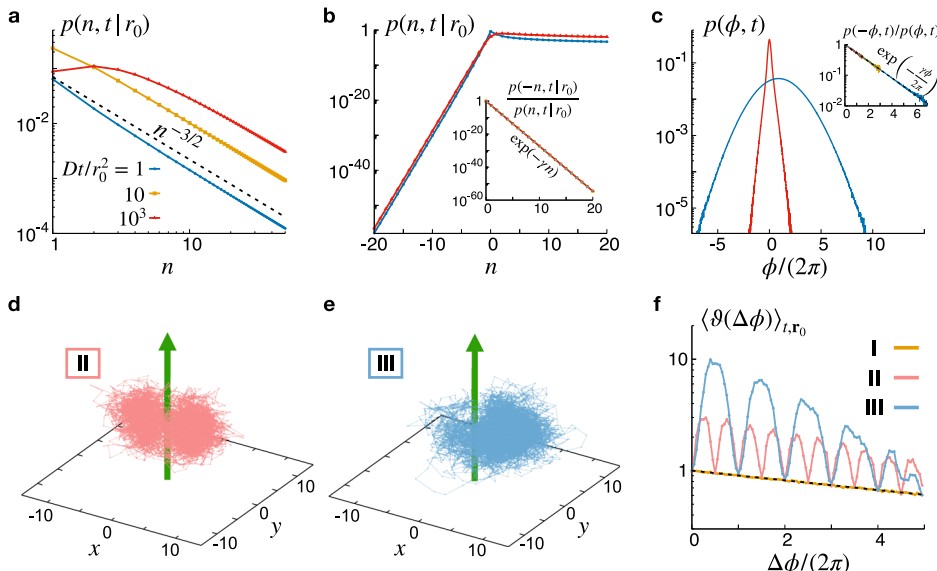

**Fig. 2 The topological fluctuation theorem for a single vortex. a** The exact winding number distribution for a free vortex exhibits a power law scaling for large positive $n$ values. **b** At all times $t$ and initial radial position $r_0$, the ratio of negative to positive winding number distributions maps onto the theoretical prediction (3) (inset) resulting in an exponential scaling of $p(n,t|r_0)$ for large negative windings (caption in panel **a**). **c** In the presence of boundaries restricting the particle motion with diffusivity $D$ inside a disk of radius $R$ and outside the vortex center, the winding angle distribution shows exponential tails at small times (red curve, $Dt/R^2 = 0.1$), while it becomes Gaussian for larger $t$ (blue curve, $Dt/R^2 = 10$). The theorem holds irrespectively of the shape of the distribution (inset, the orange curve corresponds to $Dt/R^2 = 1$). **d, e** Representative trajectories corresponding to cases II and III discussed in the text. **f** The probability ratio (14) averaged over initial positions $\mathbf{r}_0$ and trajectory times $t$ as function of the total winding angle $\Delta\phi$ for the three cases (case I corresponding to isotropic confinement) discussed in the text and $\Delta r = \Delta z = 0$. In **a, b** $\gamma = 2\pi$, while in **c–e** $\gamma = 0.1 \times 2\pi$. In **d–f** we used $k_B T/k = 10$, while for case II (resp. III) $\alpha = 0.5(1)$ and $x_c = 0(4)$.

However, from the above derivation the relation (3) holds even when the Brownian particle is allowed to move transversally to flow streamlines. Considering closed trajectories winding around a vortex line in free space, the Fokker–Planck equation describing the evolution of the distribution $\mathcal{P}(r,\phi,z,t)$ can be solved exactly[53], such that the winding distribution is given, up to a normalizing constant, by (details in the Methods section below)

$$p(n,t|r_0) \propto \int_0^\infty du\, \mathrm{Re}\left[e^{2i\pi n u} I_{k_u}\left(\frac{r_0^2}{2Dt}\right)\right], \quad (12)$$

where $k_u = \sqrt{u^2 + iu\gamma/(2\pi)}$, $r_0$ denotes the initial radial position of the particle and $I_\nu$ is the modified Bessel function of the first kind, of order $\nu$.

A detailed examination of this probability distribution reveals that it is strongly non-Gaussian. Indeed, for positive winding numbers $p(n,t|r_0)$ is asymptotically scale free: $p(n,t|r_0) \underset{n\to+\infty}{\sim} n^{-3/2}$ [53] (see Fig. 2a), while, as a consequence of the fluctuation theorem (3), it decays exponentially as $n \to -\infty$. This difference in scaling behaviors results in a strong asymmetry of the overall distribution between the positive and negative winding number sectors (see Fig. 2b).

**Computational verification of the theorem.** For more complicated force profiles, solving the Fokker–Planck equation may not be possible, and one needs to resort to Brownian dynamics simulations. Below we address two such non-trivial cases in order to illustrate the quantization of the medium entropy production in presence of an external potential, as well as the application of the theorem to the multi-vortex case.

As the data shown in Fig. 2b suggests, a numerical verification of the theorem in free space would be very demanding due to the need for excessive sampling. On the contrary, constraining the stochastic trajectories inside a finite volume with an externally

applied potential $U(\mathbf{r})$ significantly speeds up the winding number statistics convergence. In practice, however, such confinement also imposes to make sure that stochastic trajectories cannot reach the vortex cores where the winding angle is ill-defined; which we achieve by means of a stiff potential barrier (see Methods for details about numerical simulations). For simplicity, we moreover consider the case of an effectively two-dimensional vortex force such that all vortex lines are along the third spatial direction, as described by Eq. (11). For such force fields, the dissipated heat (7) reads

$$\Gamma = \sum_{i=1}^m \frac{\gamma_i \Delta\phi_i}{2\pi} - \frac{\Delta U}{k_B T}, \quad (13)$$

where, as before, $\gamma_i$ denotes the $i$th vortex strength, while $\Delta\phi_i$ and $\Delta U$ are the corresponding total winding angle and potential energy difference between the final and initial positions. Note that Eq. (13) holds irrespectively of the observation time, such that the numerical verification of the theorem can be carried out either looking at finite time distributions, or after time averaging at fixed initial conditions.

**Quantization of the medium entropy production.** We first discuss the quantization of the medium entropy production for closed trajectories as predicted by Eqs. (2) and (7) in a configuration where the Brownian particle is harmonically trapped, namely for which

$$U(x,y,z) = \frac{k}{2}\left[(x - x_c)^2 + \alpha^{-2}y^2 + z^2\right],$$

where $(x,y,z)$ denote the cartesian coordinates of $\mathbf{r}$, while the parameter $\alpha$ introduces anisotropy in the confinement and the position of the potential minimum can be shifted along $x$ by varying $x_c$. In an experiment, $U$ would, e.g., model the confinement operated on a colloidal particle by optical tweezers.

From Eq. (13), the dissipated heat only depends on the total winding angle around the vortex and the potential difference between both ends of the trajectories. It is thus enlightening to consider the probability ratio

$$\vartheta(\Delta\phi, \Delta\mathbf{r}, t|\mathbf{r}_0) \equiv \frac{p(-\Delta\phi, -\Delta\mathbf{r}, t|\mathbf{r}_0 + \Delta\mathbf{r})}{p(\Delta\phi, \Delta\mathbf{r}, t|\mathbf{r}_0)} = e^{-\Gamma}, \qquad (14)$$

where $p(\Delta\phi, \Delta\mathbf{r}, t|\mathbf{r}_0)$ denotes the joint probability of total winding angle $\Delta\phi$ and difference between initial and final positions $\Delta\mathbf{r}$ for a trajectory of length $t$ given an initial condition $\mathbf{r}_0$. Naturally, setting $\Delta\mathbf{r} = \mathbf{0}$ (closed trajectories) leads to $\Delta\phi = 2\pi n$ and vanishing $\Delta U$, such that the theorem (8) is readily recovered.

Let us now consider trajectories for which the initial and final points share the same radial and axial coordinates: $\Delta r = \Delta z = 0$, while the total winding angle $\Delta\phi$ remains arbitrary. In case I where the potential is isotropic ($\alpha = 1$) and its ground state coincides with the vortex core ($x_c = 0$), $\Delta U$ is independent of the winding angle $\Delta\phi$ such that the probability ratio $\vartheta$ averaged over trajectory time and initial positions follows the predicted exponential scaling for all winding angles (see orange line in Fig. 2f). In general, however, $\Gamma$ is affected by the symmetries of the trap such that $\vartheta$ does not decay exponentially with the winding angle $\Delta\phi$. Setting $\alpha \neq 1$ (case II, Fig. 2d), $\Delta U$ depends explicitly on $\Delta\phi$ and vanishes only for $\Delta\phi = n\pi$. Consequently, the averaged winding angle probability ratio departs from the exponential scaling for all $\Delta\phi$ not multiple of $\pi$ (see red line in Fig. 2f). Moreover, in the third case of an isotropic harmonic potential, but whose minimum is shifted of $x_c \neq 0$ (Fig. 2e), the winding angle probability ratio shows similar oscillations as in case II, except that due to the lack of reflection symmetry of $U$ those are $2\pi$-periodic (blue line in Fig. 2f). In all cases, thanks to the topological protection offered by closed loops the data shown in Fig. 2f systematically falls on the predicted exponential curve for $\Delta\phi$ integer factor of $2\pi$.

**The multi-vortex case**. To simplify the following analysis and allow for the study of fixed time trajectories, we now consider the case where the potential $U(\mathbf{r})$ acts as a stiff wall confining the particles inside a cylinder of radius $R$ and such that $U = 0$ inside of the cylinder. For such confinement and a single vortex line located at $r = 0$, we find that for $t \ll R^2/D$ the distribution $p(\phi, t)$ exhibits exponential tails, while for $t \gg R^2/D$ the effect of confining boundaries leads to $p(\phi, t)$ being Gaussian (see Fig 2c). As expected from Eq. (13), in this case one can set $\Delta U = 0$ and the fluctuation theorem is found to hold at winding angles for all observation times, independently of the particular shape of the distribution.

We now address the multi-vortex case considering a configuration of four counter rotating vortices of strengths $\pm Q$ in a closed domain of radius $R$, all located at $r = R/2$ and 90 degrees from each other, as depicted in Fig. 1d, except that for simplicity the particle motion in simulations is restricted to two dimensions.

To be able to sample winding number distributions with sufficiently accurate statistics, we define the joint probability $p(\{\phi_+, \phi_-\}, t)$ of total winding angles $\phi_+$ and $\phi_-$ around, respectively, counter-clockwise (CCW) and clockwise (CW) rotating vortices. The distribution, shown in Fig. 3a, exhibits a non-trivial, time-invariant, $2\pi$-periodic structure. In particular, while $p(\{\phi_+, \phi_-\}, t)$ shows local maxima when both $\phi_+$ and $\phi_-$ are integer factors of $2\pi$, it vanishes up to numerical precision when both of them are odd integer factors of $\pi$. Despite this complex behavior, representing the ratio $p(\{-\phi_+, -\phi_-\}, t)/p(\{\phi_+, \phi_-\}, t)$ as a function of $\gamma(\phi_+ - \phi_-)$ reveals a clear exponential scaling at all accessible times (see Fig. 3b), in agreement with Eq. (9).

We now examine the total entropy production distribution $p(\Gamma, t)$. Figure 3c shows that although it exhibits a Gaussian envelope, it features prominent modulations that can not be described in terms of simple functions. Nevertheless, the ratio $p(-\Gamma, t)/p(\Gamma, t)$ always verifies Eq. (10) at all times and for all winding angle configurations (see Fig. 3d).

Lastly, our Brownian dynamics simulations allow us to record the individual winding distributions $p_\pm(\phi, t)$ associated with CCW($+$) and CW($-$) rotating vortices. As shown in Fig. 3e, the two distributions show smooth damped oscillations and are symmetric with respect to each other. Examining the ratios $p_\pm(-\phi, t)/p_\pm(\phi, t)$, we find that they do not satisfy the theorem. Namely, we find in both cases that the probability ratio of winding oppositely to the direction set by the vortex flow to that of winding along it is always larger than predicted by the theorem (see Fig. 3f). This striking feature is a direct consequence of the fact that in presence of many vortices, trajectories winding with an angle $\phi$ around a given vortex are not all topologically equivalent.

**Towards a description of emergent topological phases**. It would be interesting to connect the fluctuation theorem (8) to the topologically protected edge currents found in a variety of condensed matter systems driven out of equilibrium[13,15–20,25]. Although in depth treatment of this problem, in particular the formulation of a counterpart of (8) for field theories describing emergent features of many body systems (see refs. [54–57] for relevant references), lies outside of the scope of this work, here we outline a possible avenue to tackle it in non-equilibrium Markov networks.

In this part, we consider the minimal model introduced and studied in ref. [20], which consists of a two-dimensional on-lattice dynamics made of repeating motifs playing the role of non-equilibrium cycles in some abstract space (conformation, chemical, etc...). We moreover restrict the study to a square lattice as represented in Fig. 4 such that to each motif, or lattice site, are associated four internal states labeled from A to D. The following results nevertheless remain easily generalizable to other types of lattices, and we show in the SM that similar conclusions can be reached from the study of an analogous one-dimensional model.

The lattice model sketched in Fig. 4 involves two kinds of transitions, respectively, referred to as "internal" and "external", respectively, between internal states (A–D) at each site or between neighboring sites. We denote $\gamma_{ext}$ and $\gamma_{in}$ the associated clockwise external and counter-clockwise internal rates, while the reverse transition rates are written with primes. The possible moves from a lattice site at position $(x, y)$ are thus:

$$
\begin{array}{ccccc}
(x, y)_D & \rightleftarrows & (x, y)_A & (x, y+a)_B & \rightleftarrows & (x+a, y+a)_C \\
\uparrow\downarrow & \gamma_{in}^{(\prime)} & \uparrow\downarrow & ; \quad \uparrow\downarrow & \gamma_{ext}^{(\prime)} & \uparrow\downarrow \\
(x, y)_C & \rightleftarrows & (x, y)_B & (x, y)_A & \rightleftarrows & (x+a, y)_D
\end{array}
$$

with $a$ denoting the lattice step. In ref. [20] it was found that for chiral systems (e.g. $\gamma_{ext}' \ll \gamma_{ext}$, $\gamma_{in}' \ll \gamma_{in}$) and fast external rates ($\gamma_{ext} \gg \gamma_{in}$) this dynamics leads to topologically protected chiral modes localized at the edge of the simulation domain.

Using a systematic coarse-graining approach, we derive the Fokker–Planck equation describing the effective dynamics of the total density $\rho \equiv \rho_A + \rho_B + \rho_C + \rho_D$:

$$\partial_t \rho = -\partial_\alpha \left[ v_\alpha \rho - \partial_\beta \left( D_{\alpha\beta} \rho \right) \right], \qquad (15)$$

where Einstein summation over repeated indices is implied. The derivation of Eq. (15) for the fully chiral case is presented in Methods, while the longer version accounting for reversed

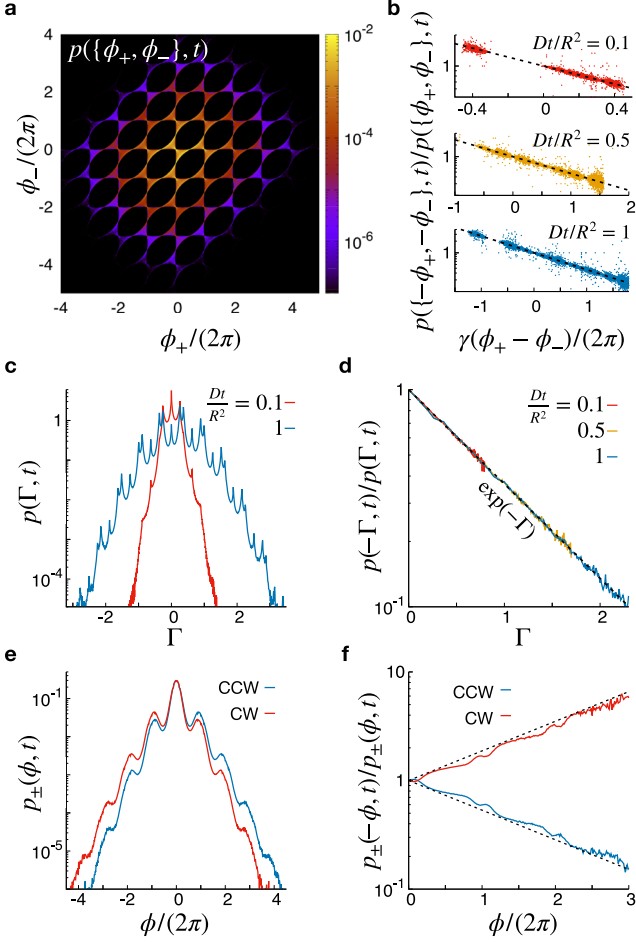

**Fig. 3 The topological fluctuation theorem in the presence of multiple vortices. a**, **b** The strong formulation of the theorem is assessed by measuring the joint CCW and CW winding angle distribution $p(\{\phi_+, \phi_-\}, t)$ (**a**) whose behavior satisfies Eq. (9) (**b**). **c**, **d** The total entropy production distribution exhibits a complex structure (**c**), but obeys the theorem nonetheless (**d**). **e**, **f** On the contrary, individual CCW(+) and CW(-) winding distributions (**e**) generally do not satisfy the theorem (**f**). In all panels $D$ denotes the particle diffusivity, $R$ is the size of the domain, while $t$ is the trajectory time. In **b**, **d**, **f** the dashed black lines indicate the theoretical exponential scaling for the probability ratios. In **a**, **e**, **f** $Dt/R^2 = 1$ and in all plots we used $\gamma = 0.1 \times 2\pi$.

transitions can be found in the SM. The effective drift **v** and diffusivity **D** are non-trivial functions of the rates $\gamma_{in}^{(\prime)}$ and $\gamma_{ext}^{(\prime)}$, and their general expressions are given in Methods. We find that the bulk dynamics is essentially diffusive with $v_\alpha^{bulk} = 0$ and $D_{\alpha\beta}^{bulk} = D_b \delta_{\alpha\beta}$. At the system's boundaries, however, $\mathbf{v}^{edge} = v_\parallel \mathbf{e}_\parallel$ is non-zero and $\mathbf{D}^{edge} = D_\parallel \mathbf{e}_\parallel \mathbf{e}_\parallel + D_\perp \mathbf{e}_\perp \mathbf{e}_\perp$, where the unit vector $\mathbf{e}_\perp$ is orthogonal to the boundaries and points inside the system, while $\mathbf{e}_\parallel$ is obtained from $\mathbf{e}_\perp$ by a $-\frac{\pi}{2}$ rotation (see Fig. 4).

Moreover, our derivation leads to conclude that the density $\rho$ takes distinct values in the bulk and at the edges of the system. Namely, we find in the limit $\gamma_{in}, \gamma'_{in} \ll \gamma_{ext}, \gamma'_{ext}$ of interest that

$$\frac{\rho^{edge}}{\rho^{bulk}} \simeq 1 + \frac{1}{4}\frac{(1 - \zeta_{in})(1 - \zeta_{ext})}{\zeta_{ext} + \zeta_{in}}, \quad (16)$$

where we have introduced the ratios $\zeta_{in} \equiv \gamma'_{in}/\gamma_{in}$ and $\zeta_{ext} \equiv \gamma'_{ext}/\gamma_{ext}$. For fully chiral rates corresponding to small or large values of the ratios $\zeta_{in}$ and $\zeta_{ext}$, it appears clearly from Eq.

(16) that most of the density is concentrated at the boundaries. Namely, in this limit $\rho$ is found to decay from $\rho^{edge}$ to $\rho^{bulk}$ exponentially with a penetration depth $\simeq a$[20]. Considering without loss of generality both $\zeta_{in} \ll 1$ and $\zeta_{ext} \ll 1$, the effective drift at the edges simplifies at leading order as $v_\parallel \simeq \gamma_{in} a$, while the bulk and edge diffusivities are given by

$$D_b \simeq \frac{\gamma_{in} a^2}{4}, \quad \frac{D_\parallel}{2D_b} \simeq 9\zeta_{ext} + \zeta_{in}, \quad \frac{D_\perp}{2D_b} \simeq \zeta_{ext} + \zeta_{in}.$$

Thus, as the system is brought to the fully chiral limit its bulk is depleted and its edges retain most of the probability density. The global dynamics is essentially one-dimensional with chiral currents traveling along the borders with mean speed $v_\parallel > 0$ and vanishing fluctuations strength $D_\parallel$.

In this context, we can apply the fluctuation theorem to this problem considering closed trajectories following the boundary of the system. We then obtain for the winding number probability ratio

$$\frac{p(-n, t)}{p(n, t)} \simeq \exp\left(-\frac{2N_{edge}}{9\zeta_{ext} + \zeta_{in}}n\right), \quad (17)$$

with $N_{edge}$ denoting the total number of lattice sites at the edge and where $n$ here corresponds to the number of times the trajectory has wound around the system. Equation (17) shows that the ratio of clockwise to counter-clockwise flux probabilities is topologically protected as it does not depend explicitly on the shape of the system boundary. However, it depends on the total perimeter as the transition rates $\gamma_{in}^{(\prime)}$ and $\gamma_{ext}^{(\prime)}$ were assumed constant. We finally note that Eq. (17) was formally obtained in the limits $\zeta_{ext} \ll 1$ and $\zeta_{in} \ll 1$. Thus and since the ratio in the exponential involves a factor that scales with the system size, computational or experimental verification of Eq. (17) might be challenging in practice.

## Discussion

We have demonstrated a topological fluctuation theorem that identifies a relevant topological invariant able to predict observable quantities, in a strongly fluctuating system without underlying periodic structure. The ratio of particle trajectories going against the flow to those going along it is purely a function of heat generated along the vortices and is topologically protected against deformations of these trajectories. Thanks to this property, the theorem holds for any finite observation time and is insensitive to conservative contributions to the force-field such as a confining potential, which makes it generic and observable in realistic experimental conditions.

In the context of micro-machines, where some degrees of freedom are driven out-of-equilibrium by external forces and torques that are even under time-reversal, the fluctuation theorem constrains the probability ratio of negative and positive entropy production created over cycles performed by the machine. Remarkably, the theorem predicts that this ratio is topologically protected for closed cycles in phase space, as discussed recently for the one-dimensional case in ref. [58], thus providing some insight for the optimization and design of micro-machines.

Finally, it will be interesting to investigate how the above results generalize with the introduction of non-equilibrium activity, be it as a correlated bath or as persistence in the particle motion[59].

## Methods

**Detailed derivation of the topological fluctuation theorem.** For closed trajectories of total duration $t$, the winding number distribution is defined up to a

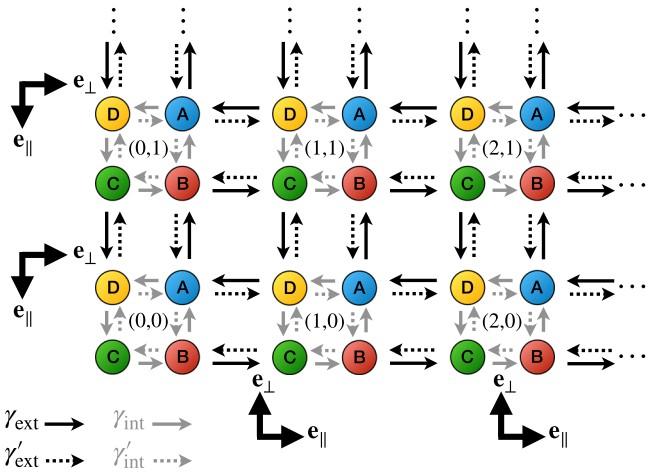

**Fig. 4 Schematic of the lattice model used to derive Eq. (15).** At each site $(x, y)$ correspond four internal states A–D while internal and external transitions are, respectively, represented by gray and black arrows. The vectors $\mathbf{e}_\parallel$ and $\mathbf{e}_\perp$ define the orientation of the system's boundaries.

normalizing constant as $p(n, t) \propto \int \mathrm{d}\mathbf{x}_0 \pi(\mathbf{x}_0) \oint_{\mathbf{x}_0} \mathcal{D}\mathbf{r}(\tau) P[\mathbf{r}(\tau)|\mathbf{x}_0] \delta(\Gamma - \gamma n)$, with $\pi(\mathbf{x}_0)$ the distribution of initial positions $\mathbf{x}_0$. Using the same convention as in the main text that $+ n$ windings correspond to trajectories evolving forward in time, we thus write

$$
\begin{aligned}
p(-n, t) &\propto \int \mathrm{d}\mathbf{x}_0 \pi(\mathbf{x}_0) \oint_{\mathbf{x}_0} \mathcal{D}\tilde{\mathbf{r}}(\tau) P[\tilde{\mathbf{r}}(\tau)|\mathbf{x}_0] \delta(\tilde{\Gamma} + \gamma n) \\
&\propto \int \mathrm{d}\mathbf{x}_0 \pi(\mathbf{x}_0) \oint_{\mathbf{x}_0} \mathcal{D}\tilde{\mathbf{r}}(\tau) P[\mathbf{r}(\tau)|\mathbf{x}_0] e^{-\Gamma} \delta(\tilde{\Gamma} + \gamma n) \\
&\propto \int \mathrm{d}\mathbf{x}_0 \pi(\mathbf{x}_0) \oint_{\mathbf{x}_0} \mathcal{D}\mathbf{r}(\tau) P[\mathbf{r}(\tau)|\mathbf{x}_0] e^{-\Gamma} \delta(\Gamma - \gamma n) \\
&= p(n, t) e^{-\gamma n},
\end{aligned}
$$

where the first equality makes use of Eq. (6), while the second equality is obtained noting that the total circulation is odd under time reversal ($\tilde{\Gamma} = -\Gamma$) and that integrating over forward and time-reversed trajectories is equivalent[42]. Finally, the the last equality derives from the relation $\Gamma = \gamma n$ valid when the rotational component of the force field satisfies Eq. (2).

**Brownian dynamics simulations.** Here, we provide the details of the Langevin dynamics simulations. Denoting $\mathbf{r}(\tau)$ the particle position at time $\tau$, which is discretized in units of $\mathrm{d}\tau$, it is updated by means of an Euler–Maruyama scheme:

$$
\mathbf{r}(\tau + \mathrm{d}\tau) - \mathbf{r}(\tau) = \mu[\mathbf{f}(\mathbf{r}(\tau)) - \nabla U(\mathbf{r}(\tau))]\mathrm{d}\tau + \sqrt{2D\mathrm{d}\tau}\, \boldsymbol{\xi}(\tau), \quad (18)
$$

where $\mathbf{f}(\mathbf{r})$ denotes the applied vortex field, $\mu$ is the mobility of the particle, $U$ is a confining potential, and $\boldsymbol{\xi}$ is a Gaussian white noise vector with unit variance.

For the multi-vortex study, the potential $U$ was chosen to ensure that all trajectories are confined inside a cylinder of radius $R$, and that they cannot reach the vortex centers where the winding angles are not defined. Namely, we used the following form for the potential

$$
U(\mathbf{r}) = \frac{k}{2} \times \begin{cases} (|\mathbf{r}| - R)^2 & |\mathbf{r}| \geq R \\ (|\mathbf{r} - \mathbf{r}_i| - R_*)^2 & |\mathbf{r} - \mathbf{r}_i| \leq R_* \\ 0 & \text{otherwise} \end{cases} \quad (19)
$$

When the length-scale $\ell \equiv \sqrt{k_B T/k}$ is small compared to $R_*$ and $R$, the trajectories do not penetrate the limiting regions with $r > R$ and $|\mathbf{r} - \mathbf{r}_i| < R_*$ (see Fig. 5), such that $\Delta U$ in Eq. (13) can be set to 0.

Rescaling space and time in Eq. (18), we set $D = 1$ and $\mu k = 10^3$. With these units, we used in all simulations $\gamma = 0.1 \times 2\pi$, $\mathrm{d}t/(\mu k) = 10^{-5}$, $R/\ell = 10^3\sqrt{10}$, and $R_*/\ell = 50\sqrt{10}$. We have verified that varying moderately these parameters did not affect our results. Both the data shown in Figs. 2 and 3 were obtained by sampling the winding distributions over $10^6$ to $10^8$ independent trajectories with uniform initial conditions.

**The two-dimensional lattice model with out-of-equilibrium cycles.** Here we detail the coarse-graining of the non-equilibrium lattice model described in the text and pictured in Fig. 4. As it involves less lengthy expressions, we restrict here to fully chiral rates ($\gamma'_\text{ext} = \gamma'_\text{in} = 0$) while for completeness the derivation for the general case is presented in the SM.

*The bulk dynamics.* We first investigate the behavior of the system far from any boundary. Assuming an infinite system or periodic boundary conditions, the dynamics each state density can be written in the following compact form

$$
\partial_t \rho_{\sigma(i)}(\mathbf{x}, t) = \gamma_\text{ext} \left[\rho_{\sigma(i-1)}(\mathbf{x} + \Delta\mathbf{x}_i, t) - \rho_{\sigma(i)}(\mathbf{x}, t)\right] + \gamma_\text{in} \left[\rho_{\sigma(i+1)}(\mathbf{x}, t) - \rho_{\sigma(i)}(\mathbf{x}, t)\right], \quad (20)
$$

where $\sigma$ is the 4-periodic map of $i \in \{0, \ldots, 3\}$ to $\{A, \ldots, D\}$, $\mathbf{x} = x\mathbf{e}_x + y\mathbf{e}_y$ and $\Delta\mathbf{x}_i = a\mathcal{R}(-\frac{i\pi}{2})\mathbf{e}_x$ where the operator $\mathcal{R}(\theta)$ rotates vectors in the plane by an angle $\theta$. We now use the fact that to Eq. (20) only corresponds one conserved quantity: the total bulk density $\rho_b = \frac{1}{4}(\rho_A + \rho_B + \rho_C + \rho_D)$. To complete the set of fields, we moreover define three additional auxiliary fields:

$$
\begin{aligned}
\rho_1 &\equiv \tfrac{1}{4}(\rho_A - \rho_B + \rho_C - \rho_D), \\
\rho_2 &\equiv \tfrac{1}{4}(\rho_A + \rho_B - \rho_C - \rho_D), \\
\rho_3 &\equiv \tfrac{1}{4}(\rho_A - \rho_B - \rho_C + \rho_D).
\end{aligned}
$$

After straightforward algebra, the set of equations given by Eq. (20) for $i = 0, \ldots, 3$ is then recast up to second order in $a$ as

$$
\partial_t \rho_b = \frac{\gamma_\text{ext}}{2} \left[\frac{a^2}{2}\Delta\rho_b - \frac{a^2}{2}\mathcal{D}\rho_1 - a\partial\rho_2 + a\bar\partial\rho_3\right], \quad (21a)
$$

$$
\partial_t \rho_1 = \frac{\gamma_\text{ext}}{2} \left[\frac{a^2}{2}\mathcal{D}\rho_b - \frac{a^2}{2}\Delta\rho_1 - a\bar\partial\rho_2 + a\partial\rho_3\right] - 2\bar\gamma\rho_1, \quad (21b)
$$

$$
\partial_t \rho_2 = \frac{\gamma_\text{ext}}{2} \left[a\bar\partial\rho_b - a\partial\rho_1 - \frac{a^2}{2}\mathcal{D}\rho_2 + \frac{a^2}{2}\Delta\rho_3\right] - \bar\gamma\rho_2 + \Delta\gamma\rho_3, \quad (21c)
$$

$$
\partial_t \rho_3 = \frac{\gamma_\text{ext}}{2} \left[a\partial\rho_b - a\bar\partial\rho_1 - \frac{a^2}{2}\Delta\rho_2 + \frac{a^2}{2}\mathcal{D}\rho_3\right] - \bar\gamma\rho_3 - \Delta\gamma\rho_2, \quad (21d)
$$

where we have defined the operators

$$
\partial \equiv \partial_x + \partial_y, \quad \bar\partial \equiv \partial_x - \partial_y, \quad \Delta \equiv \partial_{xx}^2 + \partial_{yy}^2, \quad \mathcal{D} \equiv \partial_{xx}^2 - \partial_{yy}^2,
$$

while $\bar\gamma \equiv \gamma_\text{ext} + \gamma_\text{in}$ and $\Delta\gamma \equiv \gamma_\text{ext} - \gamma_\text{in}$.

Since the total bulk density $\rho_b$ is the only slow field, we now enslave $\rho_{1,2,3}$ keeping terms up to second order in $a$. A quick inspection of the rhs of Eq. (21a) indicates that $\rho_1$ should be expressed at zeroth order in $a$, while the expressions of $\rho_2$ and $\rho_3$ should include terms up to order $a$. However, we find from Eq. (21b) that setting $\partial_t \rho_1 = 0$ the resulting solution for $\rho_1$ is at least of order $a$, such that the $\rho_1$ terms contribute to orders $a^3$ and $a^2$ in the equations for $\rho_b$ and $\rho_{2,3}$, respectively. We thus neglect these terms in what follows.

Setting $\partial_t \rho_2 = \partial_t \rho_3 = 0$, we end up with

$$
\begin{pmatrix} \bar\gamma & -\Delta\gamma \\ \Delta\gamma & \bar\gamma \end{pmatrix} \begin{pmatrix} \rho_2 \\ \rho_3 \end{pmatrix} = \frac{\gamma_\text{ext} a}{2} \begin{pmatrix} \bar\partial \\ \partial \end{pmatrix} \rho_b + \mathcal{O}(a^2). \quad (22)
$$

Inverting the $2 \times 2$ matrix on the lhs, we finally get

$$
\begin{pmatrix} \rho_2 \\ \rho_3 \end{pmatrix} = \frac{\gamma_\text{ext} a}{2(\gamma_\text{ext}^2 + \gamma_\text{in}^2)} \begin{pmatrix} \gamma_\text{ext}\partial_x - \gamma_\text{in}\partial_y \\ \gamma_\text{in}\partial_x + \gamma_\text{ext}\partial_y \end{pmatrix} \rho_b + \mathcal{O}(a^2). \quad (23)
$$

Replacing these expressions in Eq. (21a), we find that the bulk density dynamics simply amounts to isotropic diffusion:

$$
\partial_t \rho_b = D_b \Delta\rho_b, \quad \text{with} \quad D_b \equiv \frac{\gamma_\text{ext}\gamma_\text{in}(\gamma_\text{ext} + \gamma_\text{in})a^2}{4(\gamma_\text{ext}^2 + \gamma_\text{in}^2)}. \quad (24)
$$

Here, we note that the bulk diffusivity is symmetric by exchange of $\gamma_\text{ext} \leftrightarrow \gamma_\text{in}$ and is essentially controlled by the smallest transition rate, as

$$
D_b \underset{\gamma_\text{ext} \gg \gamma_\text{in}}{\sim} \frac{\gamma_\text{in}a^2}{4}, \quad \text{and} \quad D_b \underset{\gamma_\text{ext} \ll \gamma_\text{in}}{\sim} \frac{\gamma_\text{ext}a^2}{4}.
$$

*The dynamics near boundaries.* It was discovered in ref. [20] that when considered in a finite volume, the lattice model dynamics is dominated by the effect of boundaries. Namely, the model shows persistent chiral modes localized at its edges. Here, we consider an effective description where the system boundary in contact with an homogeneous bulk whose density satisfies Eq. (24). Considering without loss of generality the bottom boundary (see Fig. 4 for a schematic) such that external transitions to B and from C are suppressed, we have

$$
\partial_t \rho_B(\mathbf{x}, t) = -\gamma_\text{ext}\rho_B(\mathbf{x}, t) + \gamma_\text{in} \left[\rho_C(\mathbf{x}, t) - \rho_B(\mathbf{x}, t)\right], \quad (25a)
$$

$$
\partial_t \rho_C(\mathbf{x}, t) = \gamma_\text{ext}\rho_B(\mathbf{x} + \Delta\mathbf{x}_2, t) + \gamma_\text{in} \left[\rho_b(\mathbf{x}, t) - \rho_C(\mathbf{x}, t)\right], \quad (25b)
$$

$$
\partial_t \rho_b(\mathbf{x}, t) = D_b \Delta\rho_b(\mathbf{x}, t) + \frac{\gamma_\text{in}}{2} \left[\rho_B(\mathbf{x}, t) - \rho_b(\mathbf{x}, t)\right], \quad (25c)
$$

where $\rho_b$ here accounts effectively the states A and D close to the boundary. As before, we now define

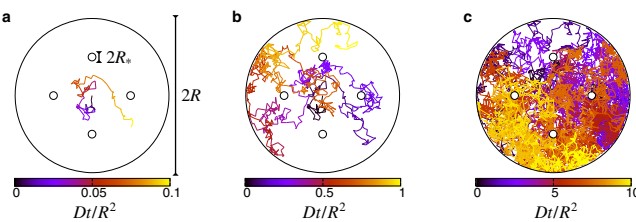

**Fig. 5 Snapshots of a simulated trajectory with four counter rotating vortices.** The trajectory starts at $r = 0$ (center) and the color labels the rescaled time $Dt/R^2$ with $D$ the particle diffusivity. When $t \ll R^2/D$ the particle has only explored a small portion of the available domain (**a**), such that it encounters the external boundaries only at larger times (**b**), while for $t \gg R^2/D$ its trajectory covers all the disk of radius $R$ (**c**). In all panels the black lines mark the boundaries set by the confining potential.

$$\rho_e \equiv \frac{1}{4}(2\rho_b + \rho_B + \rho_C),$$
$$\rho_1' \equiv \frac{1}{4}(2\rho_b + \rho_B - \rho_C),$$
$$\rho_2' \equiv \frac{1}{4}(2\rho_b - \rho_B + \rho_C).$$

Carrying out a similar calculation as for the bulk dynamics, we find that $\rho_e$ is the only conserved field such that $\rho_{1,2}'$ can be enslaved to it. Solving the resulting equations for $\rho_{1,2}'$, we get

$$\rho_1' = \frac{2\gamma_{in} - \gamma_{ext}}{4\gamma_{in} + \gamma_{ext}}\rho_e + \frac{3a\gamma_{ext}\gamma_{in}}{(4\gamma_{in} + \gamma_{ext})^2}\partial_x\rho_e + \mathcal{O}(a^2), \quad (26a)$$

$$\rho_2' = \frac{2\gamma_{in} + \gamma_{ext}}{4\gamma_{in} + \gamma_{ext}}\rho_e - \frac{a\gamma_{ext}\gamma_{in}}{(4\gamma_{in} + \gamma_{ext})^2}\partial_x\rho_e + \mathcal{O}(a^2). \quad (26b)$$

Replacing these expressions into that of $\rho_e$, we finally obtain

$$\partial_t\rho_e = -v_\parallel\partial_x\rho_e + D_\parallel\partial_{xx}^2\rho_e + D_\perp\partial_{yy}^2\rho_e, \quad (27)$$

with the coefficients

$$v_\parallel \equiv \frac{\gamma_{ext}\gamma_{in}a}{4\gamma_{in} + \gamma_{ext}},$$
$$D_\parallel \equiv \frac{\gamma_{ext}\gamma_{in}^2(5\gamma_{ext}(\gamma_{ext} + \gamma_{in}) + 8\gamma_{in}^2)}{2(4\gamma_{in} + \gamma_{ext})^2(\gamma_{ext}^2 + \gamma_{in}^2)}a^2,$$
$$D_\perp \equiv \frac{\gamma_{ext}\gamma_{in}^2(\gamma_{ext} + \gamma_{in})}{2(4\gamma_{in} + \gamma_{ext})(\gamma_{ext}^2 + \gamma_{in}^2)}a^2.$$

Finally, for a general boundary defined by the unit vectors $\mathbf{e}_\parallel$ and $\mathbf{e}_\perp$, respectively, tangent and orthogonal to the boundary, where $\mathbf{e}_\perp$ always points inside the system and $\mathbf{e}_\perp = \mathcal{R}(\frac{\pi}{2})\mathbf{e}_\parallel$, Eq. (27) is rewritten in a more general form as

$$\partial_t\rho_e = -\nabla \cdot (v_\parallel\mathbf{e}_\parallel\rho_e) + \nabla\nabla : (\mathbf{D}\rho_e), \quad (28)$$

with the Frobenius norm defined as $\mathbf{A} : \mathbf{B} = \mathrm{Tr}(\mathbf{AB})$ and $\mathbf{D} \equiv D_\parallel\mathbf{e}_\parallel\mathbf{e}_\parallel + D_\perp\mathbf{e}_\perp\mathbf{e}_\perp$.

## Data availability

The data supporting the main findings of this study are available in the paper and its Supplementary Information. Any additional data can be available from the authors upon request.

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

## Acknowledgements

We thank Viktoryia Novak for assistance with graphic design. The work has received support from the Max Planck School Matter to Life and the MaxSynBio Consortium, which are jointly funded by the Federal Ministry of Education and Research (BMBF) of Germany, and the Max Planck Society.

## Author contributions

B.M. designed and performed the simulations and carried out analytical calculations. E.T. contributed analytical calculations and participated in writing the manuscript. R.G. conceived the research, contributed analytical calculations, and supervised the project. All authors analyzed data and contributed to the redaction of the manuscript.

## Funding

## Competing interests

The authors declare no competing interests.
