## [Peer Review File · Nature Communications]

REVIEWER COMMENTS

Reviewer #1 (Remarks to the Author):

The authors proposed a "topological" fluctuation theorem, which claims that a "quantized" topological charge appears in the presence of vortices in the background force field of a Brownian particle. However, I have doubt on the validity of the main claim.

The main claim of the manuscript is Eq.(3) in which the topological charge n appears. I believe that the quantization of the charge (that "depends only by the winding number around each vortex core" as claimed in the abstract) is crucially important to ensure the novelty in this work, because without it, Eq.(3) is nothing but an ordinary fluctuation theorem.

The topological nature of heat could stem from the expression of Eq.(2), which is indeed reminiscent of a topological charge known in condensed matter physics. However, in the present setup, such a topological expression is basically not relevant to experimentally accessible quantities. In fact, Eq. (2) is based on the assumption that the trajectory of the particle is given by a closed loop, which is necessary to apply the Stokes theorem. In the present setup, however, dynamics of the Brownian particle is stochastic, so it is quite rare that the particle forms an exactly closed loop. Such a probability is characterized by the first-passage-time distribution, which is not taken into account in the present work. Even worse, the time interval of observation, t , is fixed in their setup. Then it would be almost impossible to observe an extremely rare event that the particle comes back to the original position at a fixed time.

In short, the quantization of n is not theoretically justified because it is based on a quite unrealistic assumption (and the role of such a rare probability is not taken into account). The winding number is thus not relevant to experimentally accessible quantities, and the claim "the probability is robust to local deformations of the particle trajectory" in the abstract is not justified too. If such topological natures of heat are absent, Eq.(3) is just an ordinary fluctuation theorem.

In fact, it seems that such a quantization of n is NOT at all observed in their numerical simulation. They claimed that the fluctuation theorem is always satisfied, but as mentioned above, it is just trivial. I believe that the numerical results do not support the theory (because the theory itself is not valid), and the numerically observed behaviors of the entropy production, such as the fluctuation theorem, are not new.

In summary, the present work does not establish a new contribution to this field, and the presentation about "topological" is invalid or quite misleading. I do not recommend the publication of this work.

Reviewer #2 (Remarks to the Author):

In this paper the authors derive a "topological" fluctuation theorem. The central result is based on a Helmholtz like decomposition and an application of the Stokes theorem and is a restatement of the fluctuation theorem. Specifically, the authors focus on the probability of observing a current wind around a "vortex" core and restate the fluctuation theorem. My main issue with the work (as it stands) is that it just seems like a restatement of the fluctuation theorem. It is really not clear how these ideas can be applied to gain insight not already available using the fluctuation theorem. To address this critique, the authors could try and apply this result to non-equilibrium metamaterials that are known to possess

topologically protected modes and illustrate why their ideas are useful. I will be happy to reevaluate this work at that stage.

Reviewer #3 (Remarks to the Author):

The authors demonstrate the existence of a topological fluctuation theorem (TFT) constraining the statistics of the winding number for stochastic trajectories around some vortices. For their demonstration, they consider the dynamics of Brownian particles subject to some external forces which contain a rotational component, and use some standard tools of stochastic thermodynamics. As tests of the TFT, they put forward an exact result for the winding distribution for a single vortex, and they perform numerical simulations in the case of multiple vortices.

The manuscript presents a very nice application of how stochastic thermodynamics can inform the statistics of a topological invariant in condensed matter. While topology has become increasingly important in guiding condensed matter studies in recent years, the role of fluctuations is more rarely addressed. In that respect, the manuscript is clearly a timely contribution which bridges recent progress taken from different fields, namely topology and stochastic thermodynamics. The writing is both concise and precise, allowing the reader to grasp rapidly the consequences of TFT. Importantly, the authors elegantly demonstrate the power of their framework in a minimal, yet non-trivial, example with multiple vortices. In particular, I find the discussion of the results in Fig. 3, namely how to use Eqs. (9-10), particularly enlightening.

Therefore, I strongly recommend publication of the manuscript in Nature Communications. Please find below some comments that the authors might be willing to consider to improve the manuscript.

1. When referring to original demonstrations of the fluctuation theorem (FT) in systems driven by thermal fluctuations, I believe that [J Stat Phys 95, 333 (1999)] should also be cited. Besides, it could be interesting to mention that FT is actually not limited to thermal systems, see for instance [PRL 92, 164301 (2004); PRL 95, 110202 (2005)] for applications in granular systems, and more recently [Nat Commun 8, 11 (2017); arXiv:2106.12962] in amorphous materials.

2. The geometric aspect of entropy production, namely how it relates to nonequilibrium currents, has been recently discussed in the context of nonequilibrium field theories in [arXiv:2108.13535]. Interestingly, the formalism of this work bears some similarities with the present manuscript. Note that the connection between entropy production (as a measure of irreversibility) and heat (as a measure of energy transfer) requires more care in nonequilibrium field theories than for particle-based counterparts (eg, see discussion in [PRX 11, 021057 (2021); arXiv:2104.06634]). It could be worthwhile to mention these works (maybe in the discussion section), and more generally the possible extensions of TFT to nonequilibrium field theories, as an interesting future direction to explore.

3. The derivation of TFT in Eq. (8) starting from Eq. (6) is actually (a little bit) more subtle than simply 'integrating over all initial positions'. For instance, see steps from Eq. (79) to Eq. (80) in [Rep Prog Phys 75, 126001 (2012)]. Hence, it could be useful to reformulate the paragraph above Eq. (8) (and maybe refer explicitly to [Rep Prog Phys 75, 126001 (2012)]) to avoid any confusion from a reader not familiar with tools of stochastic thermodynamics.

Point-by-point response to the reviewers' comments

Below we provide detailed answers to the reviewers' comments. The reviewers' reports are reproduced verbatim and appear in black, while our responses are shown in blue.

Reply to reviewer #1

The authors proposed a "topological" fluctuation theorem, which claims that a "quantized" topological charge appears in the presence of vortices in the background force field of a Brownian particle. However, I have doubt on the validity of the main claim.

The main claim of the manuscript is Eq.(3) in which the topological charge n appears. I believe that the quantization of the charge (that "depends only by the winding number around each vortex core" as claimed in the abstract) is crucially important to ensure the novelty in this work, because without it, Eq.(3) is nothing but an ordinary fluctuation theorem.

The topological nature of heat could stem from the expression of Eq.(2), which is indeed reminiscent of a topological charge known in condensed matter physics. However, in the present setup, such a topological expression is basically not relevant to experimentally accessible quantities. In fact, Eq. (2) is based on the assumption that the trajectory of the particle is given by a closed loop, which is necessary to apply the Stokes theorem. In the present setup, however, dynamics of the Brownian particle is stochastic, so it is quite rare that the particle forms an exactly closed loop. Such a probability is characterized by the first-passage-time distribution, which is not taken into account in the present work. Even worse, the time interval of observation, t , is fixed in their setup. Then it would be almost impossible to observe an extremely rare event that the particle comes back to the original position at a fixed time.

In short, the quantization of n is not theoretically justified because it is based on a quite unrealistic assumption (and the role of such a rare probability is not taken into account). The winding number is thus not relevant to experimentally accessible quantities, and the claim "the probability is robust to local deformations of the particle trajectory" in the abstract is not justified too. If such topological natures of heat are absent, Eq.(3) is just an ordinary fluctuation theorem.

In fact, it seems that such a quantization of n is NOT at all observed in their numerical simulation. They claimed that the fluctuation theorem is always satisfied, but as mentioned above, it is just trivial. I believe that the numerical results do not support the theory (because the theory itself is not valid), and the numerically observed behaviors of the entropy production, such as the fluctuation theorem, are not new.

In summary, the present work does not establish a new contribution to this field, and the presentation about "topological" is invalid or quite misleading. I do not recommend the publication of this work.

We thank the reviewer for their comments, which motivated the addition of new material that, we hope, will convince them of the topological nature of the theorem and that quantization of entropy production is in fact an observable quantity.

To address this concern, the section of the manuscript presenting numerical results and Fig. 2 has been extended to include a discussion about how quantization of entropy production can be measured in practice (see the section starting with "Quantization of the medium entropy production" together with panels (d-f) of Fig. 2). There, we present new simulations in the presence of a harmonic potential, which could model a colloidal particle trapped by optical tweezers. While we agree with the

reviewer that in free space the probability to observe a trajectory closing on itself is very small, this is not the case when the particle is confined within a finite volume. Moreover, although for presentation purposes the topological fluctuation theorem is derived from finite time trajectories, in the setup of interest the medium entropy production is independent of the observation time; an important feature that we would like to emphasize. In this case, Eq. (3) can be straightforwardly averaged over time so that the theorem also holds for time-averaged distributions. All these observations are now summarized in Fig. 2f where it is shown that the medium entropy production as function of the winding angle around a single vortex line generally depends on the symmetries and strength of the trap, except for closed loops where it falls on the predicted exponential curve, showing a discrete signature.

With these additions, we hope the reviewer will be able to appreciate the novelty and experimental relevance of our results.

Reply to reviewer #2

In this paper the authors derive a “topological” fluctuation theorem. The central result is based on a Helmholtz like decomposition and an application of the Stokes theorem and is a restatement of the fluctuation theorem. Specifically, the authors focus on the probability of observing a current wind around a “vortex” core and restate the fluctuation theorem. My main issue with the work (as it stands) is that it just seems like a restatement of the fluctuation theorem. It is really not clear how these ideas can be applied to gain insight not already available using the fluctuation theorem. To address this critique, the authors could try and apply this result to non-equilibrium metamaterials that are known to possess topologically protected modes and illustrate why their ideas are useful. I will be happy to reevaluate this work at that stage.

We thank the reviewer for this very interesting suggestion. Topologically protected modes are certainly a striking example of currents that form closed loops around the sample edge and hence are relevant to our framework. While bridging the topological protection arising in nonequilibrium metamaterials (that is an emergent property of a many-body system) and that occurring in the vortex-driven Brownian motion considered here (that is a single-particle non-equilibrium driven system) is a relevant question, a comprehensive answer clearly lies outside the scope of this work. We are sure the referee would agree with us that such applications of the fluctuation theorem to many-body systems have not been attempted in more than a handful of papers in a field where tens of thousands of papers have been published in the last few years.

Nevertheless, using coarse-graining techniques to map on-lattice dynamics to a Fokker-Planck description we show how to address this point in a new section entitled “Towards a description of emergent topological phases”. There, we study a lattice model introduced in [Tang et al. PRX 2021] and comprising internal chiral cycles ruling the evolution of the external dynamics between lattice sites (see the new Fig. 4). In the limit of fast external transition rates, this nonequilibrium dynamics has indeed been shown to lead to topologically protected chiral modes localized at the system’s boundary.

Using standard coarse-graining techniques detailed in Methods, we have derived the Fokker Planck equation (15) which maps this problem to the motion of an overdamped driven Brownian particle. Looking at the expression of the equation’s coefficients in the regime of interest, we recover the findings of Tang et al. that the state’s probability density is localized at the edges of the system, as well as the presence of nonequilibrium chiral currents that form closed loops around the sample edge. Our stochastic description moreover allows for current fluctuations against the direction set by the bulk chirality, so that the corresponding winding number probability ratio obeys a fluctuation theorem

(Eq. (16)) similar to Eq. (3). For uniform transition rates, however, the factor in the exponential takes a system size dependency so that such fluctuations would be hardly observable in practice.

In the Supplementary Material, we moreover show that similar results can be derived in one dimension.

To complete our response, we moreover stress that, contrary to the usual detailed fluctuation theorems, Eq. (3) is valid for any initial distribution of particles, and therefore, it does not require a steady state assumption. This feature follows from the fact that the dissipated heat Γ along a closed loop is a topological invariant only depending on the winding number, as described in the paragraph below Eq. (7) and now demonstrated numerically in the new section “Quantization of the medium entropy production”. We believe that this constitutes an important conceptual difference with the fluctuation theorem.

Reply to reviewer #3

The authors demonstrate the existence of a topological fluctuation theorem (TFT) constraining the statistics of the winding number for stochastic trajectories around some vortices. For their demonstration, they consider the dynamics of Brownian particles subject to some external forces which contain a rotational component, and use some standard tools of stochastic thermodynamics. As tests of the TFT, they put forward an exact result for the winding distribution for a single vortex, and they perform numerical simulations in the case of multiple vortices.

The manuscript presents a very nice application of how stochastic thermodynamics can inform the statistics of a topological invariant in condensed matter. While topology has become increasingly important in guiding condensed matter studies in recent years, the role of fluctuations is more rarely addressed. In that respect, the manuscript is clearly a timely contribution which bridges recent progress taken from different fields, namely topology and stochastic thermodynamics. The writing is both concise and precise, allowing the reader to grasp rapidly the consequences of TFT. Importantly, the authors elegantly demonstrate the power of their framework in a minimal, yet non-trivial, example with multiple vortices. In particular, I find the discussion of the results in Fig. 3, namely how to use Eqs. (9-10), particularly enlightening.

Therefore, I strongly recommend publication of the manuscript in Nature Communications. Please find below some comments that the authors might be willing to consider to improve the manuscript.

We warmly thank the reviewer for their appreciation of our work and recommending it for publication. Below we provide detailed answers to the reviewer’s comments.

1. When referring to original demonstrations of the fluctuation theorem (FT) in systems driven by thermal fluctuations, I believe that [J Stat Phys 95, 333 (1999)] should also be cited. Besides, it could be interesting to mention that FT is actually not limited to thermal systems, see for instance [PRL 92, 164301 (2004); PRL 95, 110202 (2005)] for applications in granular systems, and more recently [Nat Commun 8, 11 (2017); arXiv:2106.12962] in amorphous materials.

We thank the reviewer for suggesting these references. All have been added in the introduction, see in the first column of the first page after the sentences starting with “The first theorem was discovered...” and “Additional relations were moreover found...”.

2. The geometric aspect of entropy production, namely how it relates to nonequilibrium currents, has been recently discussed in the context of nonequilibrium field theories in [arXiv:2108.13535].

Interestingly, the formalism of this work bears some similarities with the present manuscript. Note that the connection between entropy production (as a measure of irreversibility) and heat (as a measure of energy transfer) requires more care in nonequilibrium field theories than for particle-based counterparts (eg, see discussion in [PRX 11, 021057 (2021); arXiv:2104.06634]). It could be worthwhile to mention these works (maybe in the discussion section), and more generally the possible extensions of TFT to nonequilibrium field theories, as an interesting future direction to explore.

We agree with the reviewer that extending the TFT to stochastic field theories is an interesting direction and we thank them for the suggestion. Unfortunately, due to the many additions to the manuscript following the revision, room for a proper description of this point is missing. We therefore chose to mention it at the beginning of the new section “Towards a description of emergent topological phases” and cited relevant references in the footnote [55].

3. The derivation of TFT in Eq. (8) starting from Eq. (6) is actually (a little bit) more subtle than simply 'integrating over all initial positions'. For instance, see steps from Eq. (79) to Eq. (80) in [Rep Prog Phys 75, 126001 (2012)]. Hence, it could be useful to reformulate the paragraph above Eq. (8) (and maybe refer explicitly to [Rep Prog Phys 75, 126001 (2012)]) to avoid any confusion from a reader not familiar with tools of stochastic thermodynamics.

We thank the reviewer for pointing this out. Indeed, the initial description of the derivation was not sufficiently detailed. We corrected this by adding some calculation steps in the Methods section (see the section “Detailed derivation of the topological fluctuation theorem”) and a reference to [Rep Prog Phys 75, 126001 (2012)].

REVIEWER COMMENTS

Reviewer #1 (Remarks to the Author):

Although the authors have provided additional numerical results, I'm still not convinced with the main claim of this manuscript. The critical point raised in my previous report has not been addressed by the authors in the revised manuscript.

An important point is that the "topological" fluctuation theorem Eq.(3) is valid only for specific events that satisfy the condition that a trajectory forms a closed loop. Under this condition, Eq.(3) is rather trivial, because it is almost equivalent to the ordinary fluctuation theorem, and moreover, quantization of n is also just trivial, because n is automatically an integer by definition. Without such special condition, on the other hand, one cannot see any quantization of thermodynamic quantities, as clearly shown in the numerical results.

Furthermore, $\gamma = Q / kT$ is not given by a universal constant, which is very contrastive to condensed matter physics (like quantum Hall effect). I agree that Eq.(2) is a kind of geometrical expression, but "geometrical" does not necessarily imply "topological", as is well understood in mathematics and condensed matter physics. In quantum materials, for example, the topological nature stems from the phase consistency condition, which ensures that the integral of curvature becomes an integer. However, such topological nature is not present in the formulation of this manuscript, but the closed-loop condition is put "by hand". I therefore consider that there is fundamental misconception of "topology" in the present work.

In summary, I believe that Eq.(3) is almost trivial by setting, and its relevance to "topology" might be conceptually wrong. I cannot recommend the publication of this work.

Reviewer #2 (Remarks to the Author):

The authors have tried to address my concerns by providing an example of a lattice based model. However, I still have some concerns about the generality/applicability of their result. For example, Eq 16 seems to be a result of assuming that the edge flux in their model is due a biased random walk. Is that so ? From the current description in the text, I was unable to see how the topological fluctuation theorem applies to the lattice model.

Reviewer #3 (Remarks to the Author):

In my opinion, the authors have satisfactorily answered to the all the criticisms raised by the Referees. Hence, again, I strongly recommend publication of the manuscript in Nature Communications.

List of main changes

- The section “Towards a description of emergent topological phases” was expanded to clarify the derivation of Eq. (17) (previously Eq. (16)).
- The Supplementary Materials now includes a section about coarse-graining of the two-dimensional lattice model in presence of both forward and backward transition rates.

Below we provide detailed answers to the reviewers’ comments. The reviewers’ reports have been copied and appear in black, while our responses are shown in blue.

Reply to reviewer #1

Although the authors have provided additional numerical results, I'm still not convinced with the main claim of this manuscript. The critical point raised in my previous report has not been addressed by the authors in the revised manuscript.

In our understanding, the primary concern expressed in the reviewer’s first report was the lack of complementary evidence for the quantization of entropy production predicted by the theorem we propose (in addition to the theorem itself), as is for example stated in this paragraph:

“In short, the quantization of n is not theoretically justified because it is based on a quite unrealistic assumption (and the role of such a rare probability is not taken into account). The winding number is thus not relevant to experimentally accessible quantities, and the claim “the probability is robust to local deformations of the particle trajectory” in the abstract is not justified too. If such topological natures of heat are absent, Eq.(3) is just an ordinary fluctuation theorem.”

We believe that the complementary numerical results which were added to the manuscript (Fig. 2f) directly address this concern by demonstrating that the winding number is an accessible quantity, and that the quantization of entropy production can be measured in a minimal configuration with anisotropic confinement. We do not understand why the reviewer does not acknowledge that this response directly addresses the point raised in the previous report. Below we address the reviewer’s additional comments.

An important point is that the “topological” fluctuation theorem Eq.(3) is valid only for specific events that satisfy the condition that a trajectory forms a closed loop. Under this condition, Eq.(3) is rather trivial, because it is almost equivalent to the ordinary fluctuation theorem, and moreover, quantization of n is also just trivial, because n is automatically an integer by definition. Without such special condition, on the other hand, one cannot see any quantization of thermodynamic quantities, as clearly shown in the numerical results.

As stated in the discussion below Eq. (7) of the manuscript, the topological nature of the theorem stems from the fact that, for closed trajectories, the dissipated heat Γ depends only on the winding number and the dimensionless quantum of heat γ which is set by the vortex strength and the medium temperature. Contrary to open trajectories which can be continuously deformed to wind or not around a vortex core, for closed trajectories the winding number n is a well-defined topological invariant. Therefore, as $\Gamma = \gamma n$ the dissipated heat does not depend on the explicit shape of closed trajectories and is topologically protected.

While the winding number n is indeed an integer by construction, it is in our opinion far less trivial that the heat dissipated by the motion of a stochastic particle performing a closed loop around a vortex is independent of its trajectory shape. This topological protection, in fact, is crucial to the derivation of the fluctuation theorem (3). As a consequence, and as stated in the manuscript, Eq. (3)

is valid for arbitrary initial distribution of particles, trajectory times, and outside of steady states contrary to most detailed fluctuation theorems (see Ref. [42]).

Furthermore, $\gamma = Q / kT$ is not given by a universal constant, which is very contrastive to condensed matter physics (like quantum Hall effect).

Indeed, the quantum of dissipated heat has *a priori* no reason to be given in terms of universal constants, as the underlying Langevin dynamics (see Eq. (1)) is not. The fact that topologically protected quantities are not functions of universal constants is simply a reflection of the fact that we are not dealing with fundamental particles (such as electrons), and is commonplace for soft condensed matter and nonequilibrium classical systems; see e.g. Refs. [10,11,14-20,25]. Therefore, we disagree with the reviewer who seems to suggest that the existence of fundamental constants in a theoretical framework has some relationship with the notion of topological protection. This is simply false.

I agree that Eq.(2) is a kind of geometrical expression, but "geometrical" does not necessarily imply "topological", as is well understood in mathematics and condensed matter physics. In quantum materials, for example, the topological nature stems from the phase consistency condition, which ensures that the integral of curvature becomes an integer. However, such topological nature is not present in the formulation of this manuscript, but the closed-loop condition is put "by hand". I therefore consider that there is fundamental misconception of "topology" in the present work.

We respectfully disagree with the reviewer about the fact that our work is unrelated to topology. A vortex force field, or topological defect, is a structure leading to a singular vorticity field as presented by Eqs. (2) and (11). As we demonstrate in the manuscript (Eq. (7)), for closed trajectories this characteristic leads to the property that the medium entropy production Γ is topologically protected. We shall moreover comment on the fact that the very notion of topological invariant commands to consider closed manifolds, whether it is in momentum space like in the study of quantum systems, or in real space as done here. The integral of curvature that the reviewer mentions only becomes an integer when it is performed over a closed loop. Therefore, we find the last sentence by the reviewer concerning "fundamental misconception of topology" to be misinformed.

In summary, I believe that Eq.(3) is almost trivial by setting, and its relevance to "topology" might be conceptually wrong. I cannot recommend the publication of this work.

We hope the answers provided above will convince the reviewer of the non-trivialness of our results and of their relation to the mathematical notion of topology.

Reply to reviewer #2

The authors have tried to address my concerns by providing an example of a lattice based model. However, I still have some concerns about the generality/applicability of their result. For example, Eq 16 seems to be a result of assuming that the edge flux in their model is due a biased random walk. Is that so ? From the current description in the text, I was unable to see how the topological fluctuation theorem applies to the lattice model.

We would like to thank the reviewer for engaging with our work and the clear question. The reviewer has asked for more clarification of the new results, and in particular, whether the lattice model can generate an edge flux only when there is a biased random walk. The answer is yes; but while unbiased rates suppress the emergence of localized chiral edge modes at the system's boundary, as we show in the revised version of the manuscript the presence of reverse rates does not qualitatively affect the conclusions initially drawn from the fully chiral version of the model. Indeed, the lattice model with

rates in both directions will exhibit an edge current when the rates are such that the penetration depth of the current, from the edge area to the bulk, is comparable to a lattice base. In this case, an edge current can be meaningfully defined and shown to have topological protection, which lends itself to the possibility of formulating the topological fluctuation theorem.

To address the reviewer's remark, we have expanded the text around the derivation of Eq. (17) (previously Eq. (16)) to clarify this aspect. In order to keep the main part of the manuscript as concise as possible, we have reported the details of the new derivation for the full lattice model in the Supplementary Material.

Let us summarize here in a few sentences the chain of arguments leading to Eq. (17). The idea behind its derivation is that, as highlighted by Eq. (16), in the limits of strong chirality and fast external rates most of the density probability is trapped at the edges of the system over a length scale that is comparable with the lattice spacing (see in particular Eq. (16) and the results of Ref. [20]). In this limit, the problem is therefore quasi-one dimensional and we moreover show that the dynamics of the density obeys Eq. (15) with a drift v_{\parallel} and diffusivity D_{\parallel} . In this context, it is thus natural to consider closed trajectories following the boundaries of the system, which leads to Eq. (17) of the main text.

Reply to reviewer #3

In my opinion, the authors have satisfactorily answered to the all the criticisms raised by the Referees. Hence, again, I strongly recommend publication of the manuscript in Nature Communications.

We thank the reviewer for reiterating their support of our manuscript and recommending its acceptance for publication.

REVIEWERS' COMMENTS

Reviewer #1 (Remarks to the Author):

The main criticism raised in my previous report can be summarized as:

- (i) The topological quantization occurs only under a specific condition that a trajectory forms a closed loop;
- (ii) Under such a closed-loop condition, the main theoretical result Eq.(3) can be derived very straightforwardly (and thus looks almost trivial);
- (iii) Therefore, it does not capture a truly topological nature of stochastic systems.

About (i), I tend to agree with the author that if the particle is trapped by a potential (not in free space), it can be experimentally observed by real experiments as suggested by Fig.2. However, I'm still not convinced with the authors about (ii) and (iii). While I agree that "fundamental misconception of topology" was a little exaggerated in my previous report, I *do* believe that the depth of the present work does not reach a novel interesting feature of topology.

I would emphasize that Eq.(3) immediately follows just from the definition of the winding number, where I do not see conceptual or technical innovation. In my opinion, this is very contrastive to other important works on topology of condensed matter (including non-Hermitian/classical systems, in which the phase-consistent condition is not always required, as the authors correctly pointed out). Therefore, I do not believe that this work is significant enough to justify publication of this work in high-impact journals including Nature Communications.

Reviewer #2 (Remarks to the Author):

The authors have addressed my comments.

Reviewer #3 (Remarks to the Author):

I acknowledge the clear effort of the authors to provide detailed answers to all the points raised by the other Referees. It leads me, again, to strongly recommend publication of the manuscript in Nature Communications.

Reply to reviewer #1

The main criticism raised in my previous report can be summarized as:

- (i) The topological quantization occurs only under a specific condition that a trajectory forms a closed loop;
- (ii) Under such a closed-loop condition, the main theoretical result Eq. (3) can be derived very straightforwardly (and thus looks almost trivial);
- (iii) Therefore, it does not capture a truly topological nature of stochastic systems.

We thank the reviewer for expressing their criticisms in a clear and concise manner. In the following we provide a detailed response which we hope will convince the reviewer of the novelty and relevance of this work to the field of topology.

About (i), I tend to agree with the author that if the particle is trapped by a potential (not in free space), it can be experimentally observed by real experiments as suggested by Fig.2.

We are pleased to learn that the reviewer acknowledges this important point. We also believe that our results shown in Fig. 2 support the fact that the quantization of the medium entropy production is an observable quantity.

However, I'm still not convinced with the authors about (ii) and (iii). While I agree that "fundamental misconception of topology" was a little exaggerated in my previous report, I *do* believe that the depth of the present work does not reach a novel interesting feature of topology.

I would emphasize that Eq. (3) immediately follows just from the definition of the winding number, where I do not see conceptual or technical innovation. In my opinion, this is very contrastive to other important works on topology of condensed matter (including non-Hermitian/classical systems, in which the phase-consistent condition is not always required, as the authors correctly pointed out).

As stated in our previous reply, a central conceptual result of this work is to identify for closed trajectories in vortex force fields that the heat dissipated (which is a measure of time-reversal symmetry) by a Brownian particle can be expressed in terms of a topological invariant, namely the winding number around vortices. This feature endows it with unique properties such as the invariance by local deformation of trajectories or the fact that it satisfies a detailed fluctuation theorem even outside of nonequilibrium steady states. While the formal steps leading to the theorem (3) may seem 'straightforward' to the reviewer, in our opinion the corresponding conceptual implications are far from obvious.

In our view the relevance of topology in this work is particularly highlighted by the results shown in Figs. 2f and 3f. In Fig. 2f we clearly demonstrate that as a consequence of the topological protection of the medium entropy production the ratio of backward to forward path probabilities does not depend on the details of the confining potential so long as one considers closed loops. Furthermore, in Fig. 3f we show that in the presence of multiple vortices one generally cannot apply the theorem if one does not consider all topological features of the Brownian trajectories (here the associated winding numbers around each vortex lines).

We are pleased to learn that the referee has accepted our rebuttal of their earlier criticism regarding the phase-consistent condition being a requirement.

Therefore, I do not believe that this work is significant enough to justify publication of this work in high-impact journals including Nature Communications.

We believe that we have addressed all the issues raised by the referee and demonstrated why our work merits publication in Nature Communications, as recommended by reviewers #2 and #3. We hope that the above answers could convince reviewer #1 about the significance of this work.